# 3D Super-Resolution Nuclear Q-FISH Imaging Reveals Cell-Cycle-Related Telomere Changes

**DOI:** 10.3390/ijms25063183

**Published:** 2024-03-10

**Authors:** Tatiana V. Pochechueva, Niko Schwenzer, Tobias Kohl, Sören Brandenburg, Gesa Kaltenecker, Bernd Wollnik, Stephan E. Lehnart

**Affiliations:** 1Department of Cardiology and Pneumology, Heart Research Center Göttingen, University Medical Center Göttingen, Robert-Koch-Str. 42a, 37075 Göttingen, Germany; tatiana.pochechueva1@med.uni-goettingen.de (T.V.P.); niko.schwenzer@med.uni-goettingen.de (N.S.); tobias.kohl@med.uni-goettingen.de (T.K.); soeren.brandenburg@med.uni-goettingen.de (S.B.); 2DZHK (German Centre for Cardiovascular Research), Partner Site Göttingen, Robert-Koch-Str. 42a, 37075 Göttingen, Germany; bernd.wollnik@med.uni-goettingen.de; 3Cluster of Excellence Multiscale Bioimaging: From Molecular Machines to Networks of Excitable Cells (MBExC), University of Göttingen, Robert-Koch-Str. 40, 37075 Göttingen, Germany; 4Collaborative Research Unit SFB 1002, University of Göttingen, Robert-Koch-Str. 40, 37075 Göttingen, Germany; 5Institute of Human Genetics, University Medical Center Göttingen, Heinrich-Düker-Weg 12, 37073 Göttingen, Germany

**Keywords:** telomere Q-FISH, super-resolution STED nanoscopy, DyMIN modality, HeLa, cardiac myocyte

## Abstract

We present novel workflows for Q-FISH nanoscopy with the potential for prognostic applications and resolving novel chromatin compaction changes. DNA-fluorescence in situ hybridization (DNA-FISH) is a routine application to visualize telomeres, repetitive terminal DNA sequences, in cells and tissues. Telomere attrition is associated with inherited and acquired diseases, including cancer and cardiomyopathies, and is frequently analyzed by quantitative (Q)-FISH microscopy. Recently, nanoscopic imaging techniques have resolved individual telomere dimensions and their compaction as a prognostic marker, in part leading to conflicting conclusions still unresolved to date. Here, we developed a comprehensive Q-FISH nanoscopy workflow to assess telomeres with PNA telomere probes and 3D-Stimulated Emission Depletion (STED) microscopy combined with Dynamic Intensity Minimum (DyMIN) scanning. We achieved single-telomere resolution at high, unprecedented telomere coverage. Importantly, our approach revealed a decrease in telomere signal density during mitotic cell division compared to interphase. Innovatively expanding FISH-STED applications, we conducted double FISH targeting of both telomere- and chromosome-specific sub-telomeric regions and accomplished FISH-STED in human cardiac biopsies. In summary, this work further advanced Q-FISH nanoscopy, detected a new aspect of telomere compaction related to the cell cycle, and laid the groundwork for future applications in complex cell types such as post-mitotic neurons and muscle cells.

## 1. Introduction

Chromosome ends, intrinsically prone to DNA damage [1,2], are protected by telomeres, repetitive DNA structures stabilized by accessory packing proteins of the shelterin complex [3]. The telomeric DNA is composed of 5′-TTAGGG-3′ tandem repeat sequences of 5 to 15 kilobase pairs (kbp) [4,5] length terminating in a 3′-single-stranded overhang [6]. Over an individual’s lifetime, somatic cell telomere length diminishes, directly correlating with aging and various diseases through a multitude of distinct mechanisms [2,7]. In post-mitotic tissues like the heart, telomere dysfunction, and attrition often coincide with inflammatory processes and other health-threatening disorders [8,9]. An established method for assessing changes in telomere length is Quantitative Fluorescence In-Situ Hybridization (Q-FISH). This technique involves fluorescent labeling of telomeric DNA in fixed cells by synthetic peptide nucleic acids (PNA), followed by quantitative fluorescence microscopy [10].

Q-FISH pioneered the revelation of telomere length heterogeneity within and between clonal cells, offering a valuable tool for quantifying attrition rates [10,11]. Initially, it relied on metaphase cell spreads, a labor-intensive method requiring mitotically active cultured cells. However, subsequent studies demonstrated successful applications of Q-FISH in analyzing isolated, post-mitotic cells and tissue sections [9,12,13,14,15]. Adaptations for high-throughput screening uncovered substantial variations in telomere lengths among different cell lines and species [16,17]. Furthermore, its versatility extends to integration with immunofluorescence and DNA stains, enabling the identification of specific chromosomes, cellular markers, and signaling events [18,19].

Despite these achievements, conventional Q-FISH approaches face limitations in fully resolving and analyzing individual telomere structures, necessitating the incorporation of super-resolution techniques. 

Super-resolution STimulated Emission Depletion (STED) microscopy [20] holds tremendous potential, especially in cardiac research, as it is compatible with optically dense, isolated cardiomyocytes or myocardial tissue sections. Despite its recent initiation [21], this approach has not yet been firmly established in the telomere field and requires specific enhancements for both preclinical and clinical applications. Most Q-FISH studies, including the only available STED study [22], have intrinsic methodological limitations that can result in limited accuracy of telomere detection and hence quantification. In part, this can be attributed to telomere analysis with single but not multiple focal imaging planes, leading to low telomere detection rates and subsequent statistical inaccuracy. Moreover, existing Q-FISH studies have not explicitly addressed the actual number of PNA probe molecules hybridized to telomere DNA. Telomere length has instead been inferred from comparative calibration measurements [23] or expressed as localization counts in single-molecule microscopy [24]. Notably, some Q-FISH studies have demonstrated three-dimensional telomere imaging and thus greatly enhanced detection capability [25]. However, they have relied on diffraction-limited imaging schemes, significantly limiting their applicability to more challenging samples since closely appositional telomeres would likely remain unresolved, as highlighted in a recent study [26]. Despite the application of deconvolution procedures in these studies, the axial sectioning capability has remained an additional limit for attaining higher detection efficiencies.

In this study, innovative labeling, recording, and analysis strategies for human telomere samples have been developed. 3D STED nanoscopy has been implemented to separate individual telomeres at high resolution in three dimensions, enhancing geometrical measurements. This includes adaptive illumination, an imaging approach to minimize photobleaching effects [27]. A corresponding strategy for 3D image analysis is introduced to result in robust and automated measurements at high telomere coverage rates. 

Thereby, cell cycle chromatin dynamics were successfully analyzed in a human model system. Based on calibration measurements, the number of hybridized PNA-probe molecules was directly assessed, addressing the relationship between telomere length, compaction, and probe accessibility. Moreover, individual telomeres analyzed by STED could be assigned to specific chromosomes in situ, opening avenues to analyze individual chromosome-specific aberrations. Additionally, initial applications on paraffin-embedded human cardiac samples have been developed, extending this tool to complex cell types such as post-mitotic neurons and muscle cells.

## 2. Results

### 2.1. Development and Validation of FISH-STED Nanoscopy in Different Cell Models

For reference, the study’s workflow has been summarized as a cartoon and written methodological overview, presented as Appendix A, respectively. 

HeLa cells were employed as human model systems to establish and optimize FISH-STED nanoscopy, the regular HeLa line with 6–8 kbp telomere length and Hela 1.3, a subclone of HeLa 1.2.11 with approximately 2.5 times longer telomeres averaging 17–23 kbp in length [16,28], respectively. 

First, we employed a standard FISH protocol on HeLa 1.3 metaphase spreads, utilizing a PNA TelC probe (CCCTAA)_3_ for G-rich telomere sequence, labeled with either Abberior Star635P or Abberior StarRed fluorophore. Telomeres could be detected as expected at the ends of chromosomes while resolving nearby telomeres from sister chromatids with STED nanoscopy (Figure 1A). We thus provided proof-of-principle for both probe-specificity and STED-imaging in our preparation. From there, we confirmed FISH-STED on intact HeLa 1.3 cell preparations based on a customized hybridization protocol (Figure 1B). STED imaging separated adjacent telomeres indistinguishable in confocal or epifluorescence microscopy, returning sub-diffraction telomere spacings and diameters of typically 130 nm (Figure 1C). 

Consequently, we applied our FISH-STED settings to more complex and optically dense post-mitotic primary cells, cardiomyocytes (Figure 1D). Telomeres could be labeled and resolved by 2D-STED nanoscopy in both nuclei preparations and intact, isolated mouse ventricular cardiomyocytes. Yet variable signal intensities and spot sizes were detected, pointing toward the need for data improvement through comprehensive 3D imaging and analysis. In parallel, we tested two concentrations of the TelC probe (Figure 1B,E) for 5 nuclei, respectively. Maximum count rates per telomere were used as a final readout and showed nearly identical distributions among telomeres from 5 nuclei for both 200 nM and 2 µM probe concentrations, respectively. This outcome indicates probe binding saturation at 200 nM.

Finally, we extended our application development to novel potential translation applications. In this context, we combined subtelomeric chromosome-specific probes with our FISH-STED approach (Figure 1F). We used a green subtelomeric probe for labeling the q-arm of chromosome 13, approximately 114 kbp remote of the telomere-associated repeat. In the HeLa 1.3 cell line, the double labeling specifically identified TelC telomere FISH signals overlapping with the subtelomeric 13q probe. Here, we could identify two distinct telomeres based on the sub-diffraction resolution of the STED channel, while only one signal spot corresponding to the 13q probe could be resolved in the confocal channel. In parallel, we performed the first FISH-STED experiment in a cardiac tissue section, a paraffin-embedded slice of a small human left-ventricular biopsy taken after severe aortic valve stenosis and minimal invasive artificial valve replacement (Figure 1G). Triple-staining with TelC probes, wheat-germ agglutinin (WGA), and DAPI discriminated telomere signals in the nuclei of heart muscle cells from neighboring fibroblast nuclei, based on WGA stainings highlighting cell borders (Figure 1G). Telomere nanoscopy resolved individual nearby telomeres, as described above.

### 2.2. 3D STED Adaptive Illumination (DyMin) Improves Imaging Quality in FISH-STED Nanoscopy

#### 2.2.1. Optimization of 3D STED Settings for Super-Resolution Time-Gated STED Microscopy

In samples with hundreds of densely packed signals, an effective single-telomere resolution was attained through specialized 3D STED imaging. Thereby, telomere coverage could be maximized and quantification occurred at telomere center-planes, respectively. For this purpose, telomeres in single isolated nuclei of HeLa 1.3 cells were labeled with TelC-Star635P and imaged in 3D super-resolution using time-gated STED microscopy [29]. Axial resolution was improved by shaping the STED laser beam using a spatial light modulator [30,31]. To efficiently sample entire nuclei, a voxel size of 30 × 30 × 200 nm was selected in accordance with the expected telomere size and optical resolution. A smaller voxel size was avoided to mitigate photobleaching and keep acquisition times reasonable. 

Additionally, Dynamic Intensity Minimum (DyMIN) scanning was employed to reduce the applied dose of STED laser light [27] and thereby minimize fluorophore photobleaching during extensive 3D imaging. As an adaptive illumination technique, the DyMIN modality dynamically controls STED laser intensity to limit the application of the STED power for actual sampling only to voxels corresponding to objects. The intensity threshold applied for pixel-by-pixel probing steps preceding the actual recording was optimized for the density and brightness of telomere spots. The resulting scanning patterns (shown in Figure 2A) significantly reduced the depletion light dose to only 2% of conventional STED microscopy. The increased photon budget allows for higher imaging resolution and signal levels, crucial for the efficient and accurate analysis of telomeric features. This aligns with the relative scarcity of telomeric signals within the nuclear volume, making DyMIN an ideal application. Consequently, an optical resolution of down to 60 nm laterally and 400 nm axially was employed (Figure 2B). To confirm that photobleaching was effectively circumvented, the confocal signal intensity was compared in the central plane of a metaphase HeLa 1.3 cell before and after 3D STED-DyMIN acquisition of the nucleus. We thereby confirmed that, typically, less than 10% of the signal was lost as a result of photobleaching.

#### 2.2.2. Automated 3D Image Analysis for Telomeric Feature Extraction

To achieve robust and unbiased quantification of telomeric features, we developed an automated workflow using the software ImageJ Fiji, initially designed for confocal data [32]. Firstly, spot-like telomere signals in 3D image stacks were identified using the plugin FindFoci [33], which employs maxima-based peak detection and three-dimensional signal segmentation (Figure 3A,B). Unlike conventional thresholding, this approach is well-suited to resolving partially overlapping signals with variable intensities, as exhibited by telomeres. Subsequently, line profiles in X and Y were measured on the 3D maxima positions of individual telomere spots, effectively using the brightest signal plane obtained for each spot for quantification (Figure 3B, green squares). The obtained profiles were fitted with a Gaussian function to extract the signal amplitude and width of each spot (full-width-at-half-maximum, FWHM; Figure 3C). Based on these values, integrated signal intensity (photon counts) and signal density were calculated for all spots with sufficient fit quality (see Materials and Methods). We validated this approach on simulated data, confirming its effectiveness in resolving even high spot densities containing partial overlaps (Appendix A).

### 2.3. Mitotic State Dependent Telomere Analysis

We initially hypothesized that chromosome condensation during metaphase, when chromosomes align in the equator of the cell between the spindle poles, might lead to a significantly decreased size of telomere spots. This would translate into an increase in telomere FISH signal density, in analogy to a study reporting decreases in FISH signal density due to telomere decompaction after shelterin loss [24]. 

#### 2.3.1. Combined Q-FISH and Cell Cycle Labeling

In order to analyze telomere compaction in cycling cultured cells, we performed combined Q-FISH and cell cycle labeling. Therefore, we applied immunofluorescence (IF) to nuclear speckles, which form distinct intra-nuclear patterns for different phases of the cell cycle [34]. Thereby, telomeric DNA was labeled by PNA-FISH, along with nuclear speckles by sc35 antibodies and genomic DNA by DAPI, respectively (Figure 4A). Confocal imaging confirmed telomere signals within nuclear volumes of approximately 300 μm² cross-sectional area and 3–6 μm in height. Signals for nuclear speckles and genomic DNA were mutually exclusive, while the dynamic patterns formed by nuclear speckles corresponding to cellular mitotic states were reproduced [34] (Figure 4B). As interphase is the longest phase, most cells were unsurprisingly found in this state. Prophase and early G1 phase, a part of interphase, were not unequivocally discernible and thus excluded. The remaining metaphase, anaphase, and telophase comprise the M-phase, which features vastly increased telomere counts. 

#### 2.3.2. Efficient Telomere Counting and Reduced Q-FISH Brightness in M-Phase

3D STED-DyMIN nanoscopy was applied to dividing HeLa 1.3 cells. Nine interphase and eight M-phase cells, as identified by the nuclear speckle co-stain, were imaged and subsequently analyzed. The latter group comprised three metaphase, three anaphase, and two telophase cells. Telomere brightness and size were quantified using automated image analysis, as described above. 

Initially, the potential interference of clustering effects and telomere subpopulations on the credibility of FISH-STED nanoscopy in interphase was tested by generating a correlation matrix for telomeric features (Appendix A). Importantly, positional data were uncorrelated with structural and brightness outputs, thus verifying unbiased telomere detection and analysis.

The number of detected telomeres in M-phase cells (207.0 ± 18.7, mean ± SD, Figure 4C) was more than doubled compared to interphase cells (89.9 ± 8.3), while the relative variance between cells was low in both groups (CV < 10%). The measured mean telomere size given by the lateral FWHM of telomere signals at their center planes’ was 132 nm, and the corresponding mean volume was 1.44 × 10^−3^ μm^3^ when assuming a nearly spherical shape of telomeres as suggested earlier [24,35]. Interestingly, telomere sizes remained unchanged during the cell cycle phase (*p* = 0.41, Figure 4D). In our analysis, telomere sizes are thus unaffected by the cell cycle, including chromosome condensation in the M-phase.

In contrast, telomere peak brightness measured as spot intensity amplitudes showed a significant decrease of 18% in the M-phase compared to the interphase (geometric mean, *p* = 0.034, nested *t*-test on logarithms; Figure 5A,B). Histogram analysis confirmed the relative darkening of telomere spots in the M-phase, with an increased occurrence of spot intensities below 40 photon counts. Telomere brightness distributions in both groups were accurately modeled by lognormal functions (r² = 0.99, 0.98) using distinct parameters for each dataset (*p* < 0.001). 

Next, the interrelationship between telomere brightness and size was analyzed at the level of single telomeres (Figure 5C). The signal density, defined by the ratio of total brightness and spot volume, was reduced by 17% (*p* = 0.040), reflecting a size-independent darkening of spots in the M-phase. Notably, the nested test design revealed significant variation between cells in the same group (*p* < 0.001). We applied nested statistical testing to account for highly variable parameters among individual cells (Appendix A), offering stronger confidence than simply comparing the mean values of group-summarized data. Therefore, this statistical test scenario is reliable in the case of low sample numbers and the limited availability of primary biological material.

### 2.4. Conversion of Calibrated Telomere Brightness to Corresponding Telomere Length 

To facilitate the biological interpretation of photon count values, we conducted brightness referencing using custom-designed DNA Origami structures as calibration samples. While Q-FISH cannot directly measure absolute telomere sequence lengths, our aim was to translate telomere spot signal intensities into bound probe numbers, allowing us to determine the length of the labeled sequence. Initially, we confirmed that signal intensity scaled linearly with fluorophore numbers, enabling the molecular counting of PNA probes through brightness referencing. The number of bound probes is then a product of telomeric repeat length and PNA probe labeling efficiency. 

DNA origami structures with defined fluorophore counts, containing on average 17, 34, or 60 molecules of Star635P, were imaged and analyzed with the same parameters as used for telomere nanoscopy (Figure 6A). Integrated signal brightness was used as a read-out independent of variable emitter density between samples. Each DNA origami reference showed normally distributed brightness values (Figure 6B). Next, a linear regression of the obtained mean values to known fluorophore counts was performed (r² > 0.99; Figure 6C). By extrapolating the resulting standard curve, the average number of probe molecules per telomere was calculated. For HeLa 1.3 cells, we determined on average 146 fluorophore probes per telomere in interphase, translating to 2628 bp of labeled telomeric sequence (Figure 6D). The M-phase telomeres on average contained 191 fluorophores, corresponding to 3438 bp of probed sequence. 

In parallel, we confirmed telomere lengths within our model systems by terminal restriction fragment (TRF) analysis, as shown in Appendix A. TRF is the established standard method to assess actual telomere length [36] and confirmed the range of published telomere lengths (Appendix A) measured by TRF for regular HeLa as 6.5 kb, HeLa 1.3 (a subclone of HeLa 1.2.11) as 18.9 kb, and Hela 2 (a subclone of HeLa-S) as 2.3 kb, respectively [37,38,39,40,41].

Comparison of PNA-FISH and TRF telomer length analysis implies that only 10–20% of telomeric sequence repeats bind TelC-labels (Appendix A) in our approach. 

This finding, together with variable telomer brightness density during the cell cycle, implies that Q-FISH brightness depends both on telomere length and also on labeling efficiency modulated by local chromatin accessibility.

## 3. Discussion 

We established novel FISH workflows with respect to staining diverse samples and introduced new imaging modalities. Specifically, we successfully applied STED microscopy to enhance FISH imaging, focusing on clustered interphase telomeres in HeLa cells with varying chromosome lengths. Expanding our approach, we extended STED-FISH to encompass primary mouse cardiomyocytes and human cardiac biopsies, incorporating dual FISH for in situ chromosome specificity. This extension broadens the possibilities for translational method development and applications, particularly in non-dividing cells such as neurons and muscle cells. This holds special significance, considering the increased attention required for understanding telomere changes in these specific cell types. In the heart, telomere dysfunction causes impaired mitochondrial biogenesis, leading to severe metabolic and contractile dysfunction [9,42]. Similarly, telomere attrition has been linked to neuronal dysfunction in neurodegenerative diseases such as Alzheimer’s [43]. While telomere shortening has been demonstrated to impede the reproductive capacity of stem cells and instigate the onset of age-associated diseases, it’s essential to note that telomere dysfunction and the activation of DNA damage response (DDR) pathways can manifest independently of telomere attrition. This can occur through the loss of shelterin components, whether associated or not with chromatin decompaction [24,35,44]. Importantly, telomere (de-)compaction can be directly quantified using biosensitive structure-function imaging approaches such as STED-FISH.

We enhanced STED-FISH applications through the integration of 3D STED super-resolution microscopy in conjunction with DyMIN adaptive illumination. This innovation yielded unprecedented coverage of individual telomeres in HeLa 1.3 cells and provided highly sensitive readouts.

Regarding PNA probe concentrations, the literature displays variations in the amounts used, influenced by the optimal balance between signal intensities and background. In the Nature Protocol for cardiac Q-FISH [45], a final concentration of approximately 1.7 µM Cy3-(CCCTAA)_3_ PNA (TelC) probe was recommended. For paraffin-embedded cardiac tissues, we utilized 2 µM of the Abberior Star635P or Star Red–labeled PNA probe. In the case of metaphase spreads, HeLa cells, and isolated mouse ventricular cardiomyocytes, we achieved signal intensity saturation at 200 nM of the PNA probe and maintained this concentration. Similarly, Vancevska et al. [35] employed 100 nM of Alexa647-labeled PNA probe, while a basic Q-FISH protocol [23] suggests 1 µM of Cy3-(CCCTAA)_3_.

The STED-DyMIN modality allowed for minimization and thereby control of the level of photobleaching as a prerequisite to robust, automated image analysis. Additionally, the cell cycle was tracked by concomitant immunofluorescence staining of nuclear speckles. We prefer this approach over the use of synchronized cells since it involves less manipulation of sensitive biological material. The synchronization process can induce stress on cells, and stress responses may introduce variability in experimental outcomes. Moreover, not all cells synchronize uniformly, leading to a mix of cells in different phases of the cell cycle, and cell cycle distribution might require additional control, like flow cytometry. Given our specific research interests, it’s essential to acknowledge that cell synchronization methods were primarily designed for dividing cells, and their applicability to post-mitotic cells, like cardiomyocytes, is limited.

Consequently, the study presented here is the first to investigate telomere changes during the cell cycle using STED nanoscopy combined with Q-FISH analysis. In M-phase, our approach achieved monitoring over 200 telomeres per cell, which was significantly higher compared to all previous Q-FISH studies, including those on HeLa 1.3 cells with elongated telomeres [16,28,41]. The high number results from our increased 3D chromosome coverage and the aberrant genome of HeLa cells, which is hypertriploid with 76–80 chromosomes [46] in contrast to the 46 chromosomes in healthy human cells. Assuming the same state of polyploidy and chromosome counts for HeLa 1.3 cells, there are putatively 312 telomeres in the M-phase, of which 66% were evidently detected in our data. In interphase cells, assumed to have 156 telomeres in G1, the longest cell cycle phase, a slightly lower detection efficiency of 58% was achieved. Although telomeric clustering could potentially lead to merged spots and thus affect detection efficiencies, it was an unlikely factor in the presented data, as telomere size was precisely resolved and unchanged by the cell cycle stage (Figure 4D).

Our findings suggest that the analyzed interphase cells were likely in the G1 phase. In contrast, during the G2 phase, doubled chromatids appear, ultimately increasing telomere counts. Importantly, the consistent doubling of spot counts in the M-phase confirms the robustness of telomere detection and argues against spot merging due to spatial crowding or association. Our analysis revealed a homologous population of interphase nuclei with 90 ± 8 telomeres detected by 3D FISH-STED (mean ± SD). Hence, our analysis did not indicate widely different ploidy in HeLa 1.3 cells, with a consistent doubling of spot counts in M-phase (n = 207 ± 19), showing low relative variance between cells in both groups (CV < 10%). Therefore, we can rule out variable polyploidy as a source of error in this study. Another potential source of error could be a large fraction of interphase nuclei in G2, where telomere length measurements and telomere counts might be affected by the ongoing synthesis of new telomeric DNA. As noted above, we detected smaller variability in telomere counts in the S versus M phases, arguing against a relevant fraction of nuclei in G2. In parallel, we detected nearly equal telomere sizes (Figure 4D) in the S and M phases, arguing against the presence of potentially decompacted, enlarged G2 telomeres during synthesis. This is further supported by the poor correlation of telomere volumes to signal density (Appendix A), while a high correlation would reflect better probe binding during decompaction below. In addition, Deplanche et al. [47] showed that asynchronous HeLa cells are 79% in G1-, 17% in S, and only 14% in G2/M-phase. This might equally hold true for HeLa 1.3, though it has not been studied to our knowledge. Whether telomere elongation is a sufficient prerequisite for elongated G2 phases is likewise unknown. Therefore, we believe we would have distinguished G1 versus G2 phase nuclei. 

In our study, a cell-cycle-dependent variation of telomere size could not be reproduced, contradicting previous PNA-FISH-STORM data and the telomere decompaction model [24]. Our method reported an average telomere size of 132 nm (FWHM), corresponding to a mean volume of 1.44 × 10^−3^ μm^3^ when assuming a nearly spherical shape of telomeres, consistent with previous studies [24,35]. These values are comparable to previously reported data, despite different super-resolution techniques, e.g., PNA-FISH-STORM in HeLa cells, measuring telomere size in terms of convex hull volume (5 × 10^−3^ μm^3^, HeLa 1.3) and gyration radius (88 nm in HeLa L, a HeLa subclone with elongated telomeres). 

Bandaria et al. [24] revealed a dramatic increase in telomeric volume after the knockdown of specific shelterin proteins, termed decompaction, which coincided with local DNA-damage response (DDR) signaling. This confirmed the protective and structural functions of the shelterin complex and proposed a functional connection. Moreover, their study found the telomeric volume to be dependent on the cell cycle and proportional to telomere length, demonstrating telomere compaction in metaphase. However, later STORM studies in murine [44] and HeLa cells [35] contradicted the decompaction model, as a telomere volume increase was not detected after Shelterin knockdown, despite the appearance of DDR signals on dysfunctional telomeres. Thus, the DDR-protective function of Shelterin proteins has been confirmed and further elaborated by FISH nanoscopy, while the role and dynamics of telomeric chromatin (de-)compaction remain controversial. 

Surprisingly, a decrease in the peak and total brightness of telomere spots by 21% was observed in the M-phase without significant changes in telomere size, leading to a decrease in signal density at the single telomere level. This finding contradicts the hypothesis of an increased signal density upon chromatin compaction in metaphase. Telomere brightness, measured by the integrated signal of telomere spots, directly represents the number of FISH probes bound to telomere sequence repeats. The overall telomere sequence length was considered constant within the same population of cultured cells and the lateral size of telomere spots remained constant. Therefore, it seems plausible that the molecular accessibility of telomeric DNA for FISH probes was reduced in the M-phase of the mitotic process, thus lowering the labeling efficiency at a molecular level. This finding confirms increased methodological variability of in situ Q-FISH compared to metaphase Q-FISH [16], which can be explained by dynamic protein recruitment [48] and chromatin remodeling [49,50] happening at telomeres under the control of the cell cycle [51,52]. 

In the recent FRET-FISH study, however, the integration of oligo DNA-FISH and FRET allowed for the measurement of chromatin compaction at individual gene loci in single cells across different cell cycle phases. The study specifically targets the biophysical properties associated with the interplay of chromatin accessibility and compaction state. Importantly, the efficiency of oligo hybridization in FRET-FISH was not compromised, even when targeting the highly condensed chromatin during mitosis [53].

In contrast, our findings challenge a foundational assumption of the PNA-Q-FISH method, which relies on a linear conversion of fluorescent brightness to telomere length, assuming a constant and high labeling efficiency of PNA nucleotides. The data presented here provide compelling evidence for a low labeling efficiency ranging from 10% to 20% (Appendix A) that varied with the cell cycle. Absolute telomere lengths derived by commonplace calibrations of Q-FISH should therefore be interpreted with care, as calibration systems are inherently different from the biological samples at hand.

In conclusion, the main findings and novelties of the present work are, firstly, the evidence of low labeling efficiency and variable coverage of telomeric tandem repeats by PNA-FISH probes compared to telomere lengths confirmed by Southern blot TRF analysis. Secondly, we overcame the intrinsic technical limitations of conventional 2D approaches by utilizing 3D-STED-DyMIN Q-FISH, where individual telomeres were detected and quantified at the appropriate imaging plane, overcoming focal shift problems. The high coverage of telomeres obtained confirms the strength of our 3D imaging and analysis approach, resulting in a more reliable and accurate readout of individual telomere brightness. Additionally, STED imaging confirmed sufficient spatial telomere separation in the xy-plane. Hence, we present a novel high-resolution tool for nanometric imaging of cardiac cells and tissue samples. This allows for the future study of telomere integrity in situ with key model systems and human samples from cardiovascular and neurodegenerative diseases, potentially in the context of future therapeutic strategies.

## 4. Materials and Methods

### 4.1. Cell Culture

Human cells were cultured at 37 °C in Dulbecco’s modified Eagle’s medium (DMEM) supplemented with 10% fetal bovine serum (FCS, Gibco) and antibiotics.

### 4.2. Human Specimens

Following informed consent, small left-ventricular endomyocardial biopsies were obtained from two patients undergoing aortic valve replacement at the University Medical Center Göttingen [54]. Left-ventricular biopsies were directly fixed in 4% paraformaldehyde, embedded in paraffin, and cut into 5 μm sections for the imaging study.

### 4.3. Isolation of Cardiac Myocytes from Mouse Ventricles

Ventricular myocytes were isolated from 8–20-week-old wild-type C57BL6/N mouse hearts, as previously described [55]. Briefly, mouse hearts were excised, mounted via the aorta to a cannula connected to a modified Langendorff perfusion setup, and perfused with a modified Ca^2+^-free Tyrode buffer (120.4 mM NaCl, 14.7 mM KCl, 0.6 mM Na_2_PO_4_, 0.6 mM KH_2_PO_4_, 1.2 mM MgSO_4_, 10 mM HEPES, 4.6 mM NaHCO_3_, 30 mM Taurin, 10 mM 2,3-Butanedione monoxime, 5.5 mM Glucose, pH7.4) at 37 °C. The hearts were then enzymatically digested by adding 2 mg/mL collagenase type II and 40 µM CaCl_2_ to the Tyrode perfusion buffer for 9 min at 37 °C. The ventricles were dissected and agitated in solution to release the cardiomyocytes. Finally, the isolated ventricular cardiomyocytes were transferred into Tyrode buffer containing 10% FBS (Sigma-Aldrich, Burlington, MA, USA) to terminate the enzymatic digestion.

### 4.4. FISH Procedure

#### Sample Preparation

Metaphase spreads were prepared according to a published protocol [23]. In brief, Hela cells were subcultured 36–48 h before colcemid treatment, aiming for approximately 70% confluency at the time of treatment. Subsequently, the cells were incubated for 15 h in a regular medium containing 0.1 μg/mL of colcemid to arrest them at metaphase and harvested by trypsinization.

After hypotonic swelling (15 min, 37 °C) in prewarmed KCl solution (0.025 M sodium citrate and 0.04 M KCl), cells were fixed in cold Carnoy fixative (3:1 (*v*/*v*) methanol/acetic acid) for 10 min at RT. The fixation process was repeated twice, then cells were washed in a fixative solution, dropped onto ice-cold glass slides, and dried on top of a humidified heating block (1 min, 80 °C). 

Alternatively, HeLa cells, cultured on collagen-coated microscope slides up to 60–70% confluency, were fixed in 4% formaldehyde in PBS for 10 min at RT. The cells were then washed twice with PBS and permeabilized in Triton X-100 buffer (0.5% Triton X-100, 20 mM Hepes-KOH (pH 7.9), 50 mM NaCl, 3 mM MgCl_2_, 300 mM Sucrose) for 10 min at RT. Subsequently, they were fixed again in 4% formaldehyde in PBS for 10 min at RT. 

Mouse ventricular cardiomyocytes (freshly isolated as described below) were plated on glass slides coated with mouse laminin (BD Biosciences) and allowed to settle for 20 min at RT. Adherent cells were consequently fixed with 4% formaldehyde in PBS for 10 min at RT and processed as described above for HeLa cells.

Alternatively, nuclei were isolated from mouse ventricular cardiomyocytes using the Nuclei Isolation Kit (Nuclei EZ Prep, Sigma-Aldrich), fixed with formaldehyde, and processed as described above.

The prepared slides were utilized for FISH studies or stored at −20 °C for future use.

The slides with the attached samples were rehydrated in PBS and fixed with 4% formaldehyde in PBS. After washing, the slides were treated with 100 μL of RNAse A solution (100 µg/mL of RNAse A in 2× Saline Sodium Citrate (SSC) buffer) for 1 h at 37 °C in a moisture-sealed slide incubation chamber. Then they were washed three times in 2× SSC and once in distilled water. Subsequently, slides were immersed in 0.005% pepsin solution (0.005% pepsin in 10 mM glycine, pH 2) for 4 min at 37 °C. After washing in PBS, the slides were fixed once more in 4% formaldehyde, washed in PBS, and dehydrated at RT in an ethanol series (70%, 85%, and 100%).

Subsequently, the slides were hybridized with Abberior Star 635P- or Abberior Star Red-labeled PNA probes (TelC, 2 µM or 200 nM) in a hybridization mix (22 mM Na_2_HPO_4_, 22 mM Tris-HCl, pH 7.4, 66% formamide, 2.2×SSC, 0.11 µg/mL Salmon Sperm DNA). TelC with N-terminal Star-635P or Star Red bound via two AEEA linker elements was custom synthesized by Eurogentec (Eurogentec Ltd., Koeln, Germany). DNA was denatured for 10 min at 85 °C, and hybridization was carried out in a moisturized hybridization chamber at 4 °C overnight or at RT for three hours. 

For duplex FISH, 40 ng of Green 13q-ter subtelomeric probe (Oxford Gene Technology, CytoCell Ltd., Cambridge, UK) was added to the hybridization mix together with the TelC probe or alone as a control, and specimens were denatured and hybridized as described above. 

After hybridization, the slides were incubated twice in a washing solution (2×SSC, 0.1% Tween-20) for 15 min at 60 °C. They were then washed twice in 2× SSC at RT and once in distilled water. Slides were air-dried, protected from light, and mounted with Prolong Gold mounting media with DAPI (Invitrogen).

### 4.5. IF-FISH Procedure

HeLa 1.3 cells were plated on collagen-coated coverslips and allowed to attach for 24–36 h before fixation, resulting in 60% confluency. After washing with PBS, cells were fixed in 4% formaldehyde in PBS for 10 min at RT. Following fixation, cells were washed with PBS and permeabilized in Triton X buffer for 10 min at RT. After permeabilization, cells were washed with PBS and blocked with a blocking solution (1 mg/mL BSA, 3% goat serum, 0.1% Triton X-100, 1 mM EDTA; PBS pH 8.0) containing 100 μg/mL DNAse-free RNAse A for 30 min at 37 °C. After blocking, cells were incubated with primary antibodies (anti-SC-35, 1:200, mouse monoclonal, Abcam) for 2 h at RT in blocking solution and washed in PBS three times for 5 min. Afterwards, cells were incubated with secondary antibodies (Abberior Star-580, 1:200) in a blocking solution for 2 h at RT and washed in PBS three times for 5 min. 

For the following FISH procedure, cells were fixed once again in 4% formaldehyde in PBS for 10 min at RT, washed in PBS, and sequentially dehydrated in 70%, 85%, and 100% ethanol, respectively. DNA denaturation and subsequent hybridization were performed as described above.

### 4.6. FISH Procedure for Paraffin-Embedded Cardiac Tissues

The protocol was adapted from Sharifi-Sanjani et al. [45]. Briefly, the paraffin on the slides was melted by incubating them in a slide warmer at 65 °C for 5 min. For deparaffinization, the slides were incubated twice in fresh xylene for 7 min each at RT. Slides were then rehydrated in an ethanol series (100%, 95%, and 70%), then in distilled water at RT, followed by rinsing in 1% Tween-20 and distilled water. Afterwards, the antigen retrieval procedure was performed. Antigen retrieval was conducted by incubating the slides in a 1/10 dilution of antigen retrieval citric buffer (Sigma-Aldrich) at 95 °C for 45 min in a water bath. Subsequently, the slides were dehydrated in an ethanol series (70%, 95%, and 100%) and air-dried.

DNA denaturation and hybridization steps were carried out as described above, with the final concentration of the PNA probe set at 2 µM. The denaturation step for human samples was performed for 5 min at 85 °C. Following hybridization, the slides underwent two 15 min washes in 70% formaldehyde, 10 mM Tris-HCl, pH 7.5 at RT in the dark. After drying in the dark, the slides were mounted as described above and imaged or stored at 4 °C in the dark.

### 4.7. TRF Assay

DNA was extracted from cells using the DNeasy Blood and Tissue Kit (Qiagen) following the manufacturer’s instructions. DNA concentration and intactness were assessed using the Genomic DNA Screen Tape (Agilent Technologies) on the 2200 Tape Station (Agilent Technologies). The southern blot was performed using the TeloTAGGG Telomere Length Assay (Roche), following supplier instructions with a few changes. Then, 500 ng of DNA was digested for 5 h. The southern blot was analyzed using the software Image Quant TL (GE Healthcare, Chicago, IL, USA) and Microsoft Excel, as suggested by Lincz, et al. [56]. The mean telomere length was calculated with the formula $\mathrm{MTL}1=\sum {NI}_{i}/\sum {(NI}_{i}/{MW}_{i})$.

### 4.8. Optical Device Setup

All imaging was performed using a custom-built microscopy setup based on the Abberior RESOLFT QUAD P microscope kit. The setup uses an Olympus IX83 inverted microscope, equipped with a 100x 1.4 NA oil-immersion objective and an Abberior QUAD beam scanner. The setup was controlled by the software Abberior Imspector version 16.1.6905. Three fluorescent imaging channels were used in confocal scanning mode, herein designated as blue (405 nm excitation, 422–467 nm detection), green (594 nm excitation, 605–625 nm detection), and red (640 nm excitation, 650–720 nm detection). All lasers were pulsed and regulated by an acousto-optic modulator. The red channel was also used in STED mode, supported by a 775 nm synchronized depletion laser, which was aligned at the start of each imaging session using 40 nm red fluorescent beads (Abberior Nanoparticle Set for Expert Line 595 and 775 nm). In STED mode, a time gating of 0.5–8 μs was used. The pinhole size was set to one Airy unit in all recordings. Reported pixel intensities are photon counts detected by avalanche photodiode detectors without further processing. The intensity calibration workflow that was demonstrated is nonetheless applicable to a variety of setups that report arbitrary intensity units, as long as the photon-to-brightness response is linear, i.e., detectors are not reaching saturated conditions. This can be evaluated based on the linear fit, as shown in Figure 6C. 

Z-stacks were recorded using continuous autofocus, an operation mode used to mitigate sample drift by compensation of small stage movements along the optical axis.

#### 4.8.1. Dynamic Intensity Minimum STED Microscopy

IF-FISH slides of HeLa 1.3 cells were prepared according to the customized protocol shown below and imaged in three channels: DAPI staining in blue, nuclear speckles IF in green (immunostaining using sc35 primary antibody, Abberior Star-580 secondary antibody), and telomere FISH in the red channel. Confocal overview images containing multiple cells were recorded at 100 nm pixel size in the blue, green, and red channels. The respective total pixel dwell times were 20, 60, and 20 μs and laser powers were 25, 20, and 2%. From the overview images, the M-phase and interphase cells were identified and chosen in random order for 3D-STED-based acquisition. 

The 3D-STED imaging of single nuclei for advanced Q-FISH telomere analysis was performed with a setting of 25% 3D-STED. Images were recorded using 30 × 30 × 200 nm pixel size (x, y, z), 8% excitation intensity, and 20% STED intensity using Dynamic Intensity Minimum (DyMIN) STED mode [27]. DyMIN parameters were set as follows: CONF level and DyMIN level were set to 15 counts. The base pixel dwell time was 10 μs (5 μs × 2 line accumulations), with one step in the confocal channel, 2 steps in the probing channel, and 4 steps in the final STED channel, resulting in a dwell time of 40 μs in the recorded STED image. The Exclusion parameter was set to 10, leading to a probing step STED intensity of 6.3%, or about one-third of the final power. 

#### 4.8.2. Molecular Brightness Calibration

To retrieve fluorophore counts from fluorescent photon counts, fluorescent bead slides (Custom Brightness DNA Origami, GATTAquant) with defined fluorophore counts of 17, 34, or 60 molecules of Star-635P per bead were measured to create a calibration curve [57]. Bead samples were produced using the same mounting medium as biological samples (ProLong Gold Antifade, Thermo Fisher Scientific, Waltham, MA, USA). 

#### 4.8.3. Image Analysis

In ImageJ Fiji version 1.52s38, microscopy images were manually cropped to include single nuclei and masked by polygons following the DAPI signal to exclude background signals if necessary. Display range and color maps were adjusted to validate the fitting process and present the data. The raw intensity values were not changed by these operations, and files were saved in the uncompressed 16-bit “.tif”-format. 

In Q-FISH analysis, TelC-Star635P signals were smoothed by a Gaussian filter and then subjected to a 3D spot detection routine based on the ImageJ plugin software FindFoci, implemented within a custom macro workflow [33]. A background parameter of 10 photon counts and a minimum spot size of 10 voxels were determined empirically using the provided graphical user interface. The resulting center of mass coordinates for each telomere spot were used to center two orthogonal lateral fits of the signal by a Gaussian function using the ImageJ curve fitting tool. The Gaussian function in ImageJ is defined as follows:$$y=a+\left(b-a\right)*e^{-\frac{(x-c)²}{2*d^{2}}}$$
in which *a* is the y-axis-offset or background level, *b* is the signal amplitude, *c* is the x-axis-offset and *d* is the standard deviation (σ). A line length of 480 nm (16 px) was used in STED images. The fit with higher goodness (r²) was used to retrieve telomere brightness and lateral size by the following equations when at least one fit was sufficient (r² > 0.8):Brightness=b
$$FWHM=2*\sqrt{2*\log2}*d$$

Volume was approximated by a 3D Ellipsoid, in which radius was defined as half of FWHM and was interpolated in z by the average of x and y because lateral resolution was much higher than axial resolution. If one fit was insufficient, only the other fit was used.
$$Volume=\frac{4\pi }{3}*r_{x}{*r}_{y}*r_{z}=\frac{\pi }{6}*{FWHM}_{x}*{FWHM}_{y}*{FWHM}_{\bar{xy}}$$

Further spot statistics were only determined when both fits were sufficient. Integrated brightness was calculated using the 2D Gaussian integral of a spot. Signal density was calculated as the ratio of integrated brightness and volume.
(1)$$Integrated\;Brightness=2\pi *b_{\bar{xy}}*d_{x}*d_{y}$$
(2)$$Signal\;Density=\frac{Integrated\;Brightness}{Volume}$$

### 4.9. Data Handling and Statistics

Data visualization and statistical analysis were performed in GraphPad Prism version 8.3.0. Scatter plots show raw data retrieved by automated spot analysis, with red lines indicating the median and interquartile range. Larger datasets, such as combined data from multiple cells, were displayed as box plots showing the median and interquartile range, with whiskers denoting the 5th and 95th percentiles. Bar graphs displayed the average n (telomere number per cell) ± SD. 

Statistical significance was determined using a nested/hierarchical one-way ANOVA or *t*-test. In this test design, cell origin or cell cycle state were used as higher-order groups and single cells as subgroups. Combining the data of multiple cells in one group and then using standard tests is common practice even in Q-FISH studies, but results in pseudo-replicates and thus falsely low *p*-values [58], since there may be high variability between cells, meaning that data points are not truly independent. The nested tests, however, can account for unequal sample sizes and variance both across subgroup data and subgroup means, which are accounted for in the group comparison. Since the test assumes a Gaussian population, data following lognormal distributions, such as brightness values, were transformed in Prism by the formula Y = log_10_(X) prior to statistical testing. 

Here, *p* values below 0.05 were defined as significant and annotated in graphs as follows: 

ns = *p* > 0.05, * = *p* ≤ 0.05, ** = *p* ≤ 0.01, *** = *p* ≤ 0.001

Frequency distribution data were fit by Gaussian curves of the following formula:$$Y=A*\exp\left(-0.5*{\left(\frac{X-µ}{\sigma }\right)}^{2}\right)$$
using robust regression and the retrieved μ and σ values used as the sample mean and SD. Lognormal distributions were similarly modeled based on the following equation:$$Y=\left(\frac{A}{X}\right)*\exp\left(-0.5*{\left(\frac{\ln\left(\frac{X}{GeoMean}\right)}{\ln\left(GeoSD\right)}\right)}^{2}\right)$$
using least-squares regression. Multiple fits were compared using the extra-sum-of-squares F test, testing if one model can fit all datasets.

## Figures and Tables

**Figure 1 ijms-25-03183-f001:**
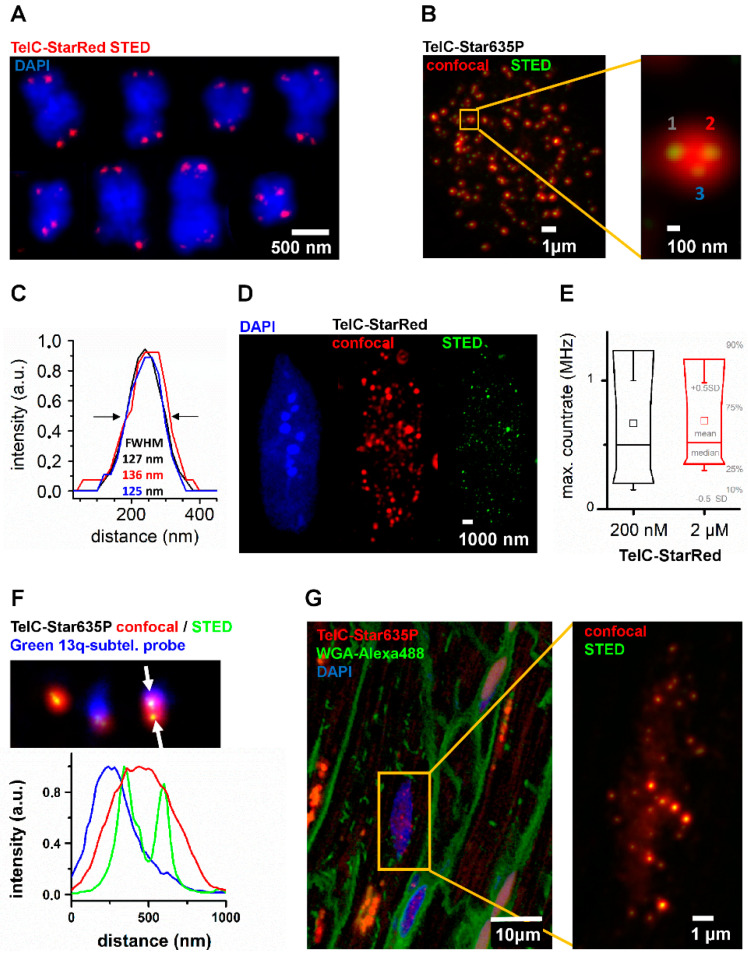
FISH-STED was employed for novel applications with potential for translational research. FISH-STED with novel TelC-StarRed and TelC-Star635P probes for STED nanoscopy was established for a variety of samples. (**A**) FISH-STED confirmed specific DNA labeling with TelC-StarRed (red signals) as shown for selected chromosomes from a HeLa 1.3 cell karyogram (blue signals, DAPI). Telomeres of aligned sister chromatids could be resolved at sub-diffraction resolution. (**B**) Analogously, FISH-STED with TelC-Star635 was performed in situ on isolated nuclei and detected spots of similar size and brightness (green) within confocal signal spots (red). The magnification (orange box) highlights how STED nanoscopy (green signals) resolved individual telomeres. (**C**) Line profiles of telomere spots 1 to 3 highlighted in (**B**) confirm nanoscopic resolution and typical telomere sizes around 130 nm of diameter determined as FWHM. (**D**) FISH-STED was also established within completely isolated mouse ventricular myocytes (VMs), a highly differentiated post-mitotic cell type. (**E**) FISH-STED signals in mouse ventricular myocytes were stable for different TelC probe concentrations as judged from maximum signal count rates per telomere spot. This indicated saturated TelC-probe binding at 200 nM (204 spots from 3 nuclei; 2 µM: 172 spots from 2 nuclei). (**F**) Individual telomeres can be assigned to specific chromosomes in situ. Hela 1.3 cells were labeled in situ with TelC-Star635P (confocal: red; STED: green) and a 13q-subtelomeric probe for chromosome 13 (blue), respectively. This double labeling identified telomere FISH-STED signals on chromosome 13, identified as a blue spot in the confocal detection channel, as highlighted by overlapping line profiles (from between arrows). (**G**) FISH-STED can be performed in human histology sections, addressing distinct cell types in triple stains. In confocal images, the nuclei of cardiomyocytes and fibroblasts could be distinguished based on the WGA-Alexa488 (green) labeling of cell membranes at the lateral and longitudinal cell borders, as well as components of the intracellular transverse-axial tubule membrane network of cardiomyocytes. FISH-STED (magnification from the orange box) resolves individual telomeres for further analysis.

**Figure 2 ijms-25-03183-f002:**
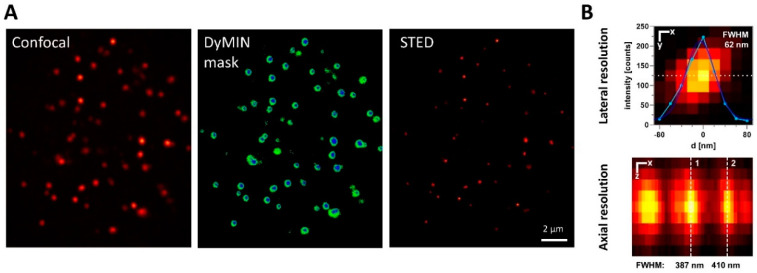
Adaptive illumination supports 3D-STED FISH Nanoscopy. (**A**) In order to reduce the bleaching of telomere signals within nuclei, adaptive illumination was applied. For positions reaching sufficient confocal signal levels (left image), low-dose STED signals were measured (center: green areas) as a probing step. Only if the resulting signal again reached a threshold level was the final STED power level applied for image generation (center: blue areas). As a result, telomere spots were resolved at super-resolution while minimizing sample light dosage when imaging in multiple planes (right). (**B**) 3D STED resolution was evaluated using 40 nm fluorescent beads as a test sample. FWHM values indicate approximately the achieved optical resolution in lateral and axial imaging. Scale bars represent 20 nm for the upper image and 100 nm for the lower image. FWHM was calculated from the Gaussian fit standard deviation of intensity line profiles along the dashed lines.

**Figure 3 ijms-25-03183-f003:**
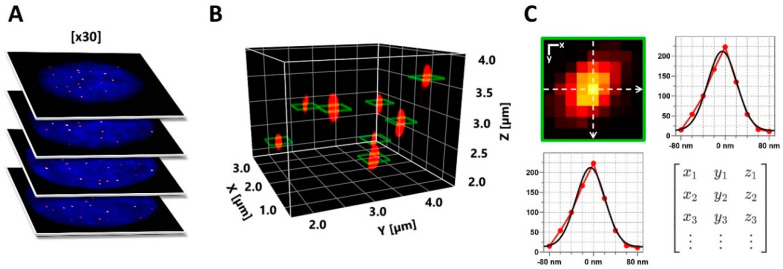
Comprehensive 3D FISH-STED imaging with adaptive illumination extracts telomeric features. (**A**) Nuclei were sampled employing increased axial resolution with 3D STED and adaptive illumination at an axial step size of Δz = 200 nm. (**B**) Next, telomere spots were reconstructed in 3D, and centroid positions were defined. (**C**) At each centroid position, telomere parameters were derived from the Gaussian fitting of two orthogonal intensity profiles in the lateral plane, offering optimal resolution. Retrieved parameters were used to calculate telomere spot peak brightness, integrated brightness, spot FWHM, volume, signal density, background level, and goodness of fit.

**Figure 4 ijms-25-03183-f004:**
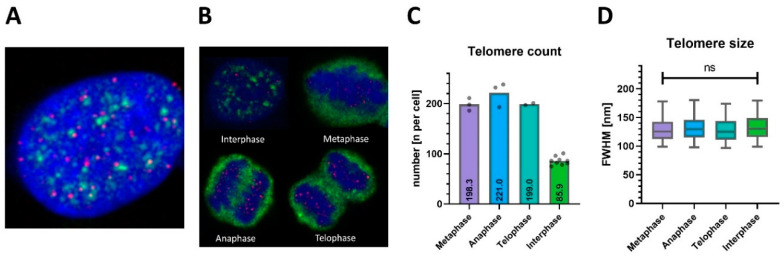
Comprehensive 3D FISH-STED nanoscopy achieves telomere monitoring throughout the cell cycle. (**A**) Telomere FISH (red) was combined successfully with immunofluorescence against spliceosome component sc35 (green), an established marker for nuclear speckles and cell cycle phase indicator. (**B**) sc35 redistributes specifically during the cell cycle. Four typical cells and their nuclei (blue) within a growing HeLa 1.3 cell culture are shown. Cells were assigned accurately to mitotic phases based on combined telomere and sc35 signal patterns. (**C**) 3D FISH-STED detected a consistent doubling of spot counts in M-phase (n = 207 ± 19) versus interphase (n = 90 ± 8) at an estimated telomere coverage of 66% in M-phase and 58% in interphase. (**D**) Telomere sizes were constant during the cell cycle, regardless of telomere counts and crowding. ns: not significant.

**Figure 5 ijms-25-03183-f005:**
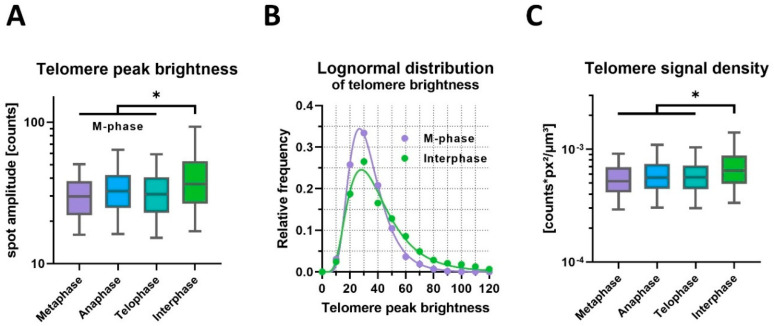
Telomere brightness varies specifically during the cell cycle. (**A**) Comprehensive 3D FISH-STED detected brighter telomeres during interphase as compared to the mitotic phase. (**B**) Telomere brightness follows a lognormal distribution. The number of darker telomeres is increased during the M-phase at the cost of the number of brighter telomeres. Interestingly, the modal peak brightness is similar in the M-phase and interphase. (**C**) Telomere signal density calculated from telomere size (FWHM) and integrated signal density increases significantly in interphase. * *p* ≤ 0.05.

**Figure 6 ijms-25-03183-f006:**
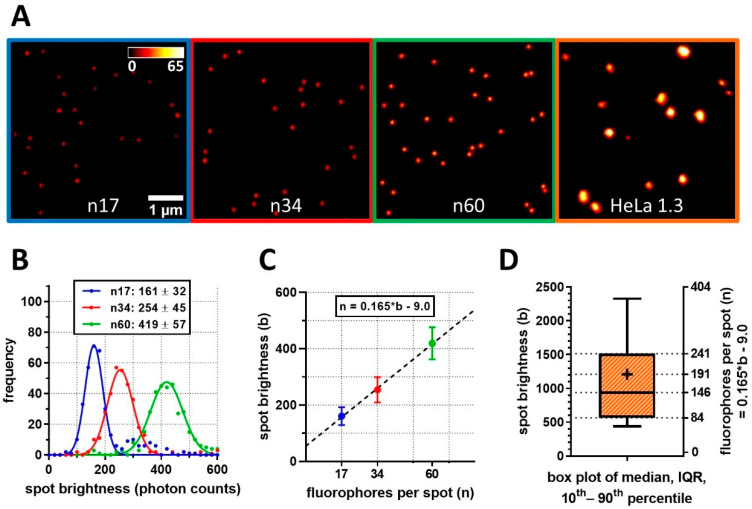
FISH-STED determines TelC probe molecule counts per telomere based on calibration measurements. (**A**) Imaging of fluorescent DNA Origami structures with defined Abberior Star635P fluorophore was performed with standardized 3D FISH-STED settings applied to in situ telomeres. (**B**) The brightness of beads measured by automated spot analysis was stochastically defined for each type based on the spot brightness distribution and fit by a normal distribution to retrieve mean and SD values. (**C**) A factor of 0.165 was determined for the conversion of spot brightness to molecular probe numbers using linear regression. (**D**) Based on the previous brightness calibration, fluorescent probe counting per telomere spot was performed and translated into the average length of the probe-labeled sequence. In interphase, median FISH-STED spot brightness corresponded to 2.63 kbp of labeled sequence stretches. Error bars indicate SD.

## Data Availability

The data are available by contacting the corresponding author.

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
