# Peer review of "3D Super-Resolution Nuclear Q-FISH Imaging Reveals Cell-Cycle-Related Telomere Changes"

_ijms, 2024, doi:10.3390/ijms25063183_

Round 1

Reviewer 1 Report

Comments and Suggestions for Authors

Super-resolution microscopy is a powerful tool for cell biology studies, especially for the visualization and quantification of small cellular structures. In this manuscript, Pochechueva et al. utilized STED microscopy to quantitatively study telomeres. By combining quantitative FISH with different modes of STED, they were able to visualize and quantify telomere number and size with single-telomere resolution. This workflow successfully demonstrated changes in telomere signal during mitosis. Additionally, they presented the capabilities of double FISH-STED and imaging in tissue. Overall, this comprehensive and useful work will benefit both the imaging and cell biology fields. I have a few suggestions, as detailed below.

1. I suggest that the authors include a graphical workflow in the main figure to enhance understanding of the entire experimental and statistical procedures.

2. Although the methodology of the current manuscript is already comprehensive, I recommend that the authors include a supplementary detailed step-by-step protocol. This will aid readers in better understanding and replicating this workflow.

3. Regarding your data, how is the photon count determined? Could you include a description in the Methods section to explain how imaging signals are converted into photon counts?

4. On Line 158, is this intended to be a subtitle?

5. On Line 162, there is additional spacing before and after “single isolated nuclei of.”

6. On Line 227, I do not think an additional subtitle here is necessary.

7. Lines 371-377 feature a different font color.

Author Response

1) I suggest that the authors include a graphical workflow in the main figure to enhance understanding of the entire experimental and statistical procedures.

Thank you for your suggestion. We appreciate your acknowledgment of the complexity of our study, involving diverse biological samples, FISH and IF-FISH protocols, microscopy strategies, and analysis. Recognizing the potential benefit for the reader and following your suggestion, we have created a graphical workflow chart to address this. We propose to include it as the first Supplementary Figure (Fig.S1), rather than in the first main figure, to avoid redundancy of the main manuscript. From there the manuscript provides essential information for what has been done in detail within each chapter, summing up to a comprehensive understanding of the entire workflow and related procedures.

2) Although the methodology of the current manuscript is already comprehensive, I recommend that the authors include a supplementary detailed step-by-step protocol. This will aid readers in better understanding and replicating this workflow.

Thank you for your recommendation that we discussed thoroughly. The graphical workflow chart presented above illustrates the distinct streams of analysis that the MS is based upon. In addition we prepared the Supplementary Protocol 1, which includes details like the applied software tools. We believe that this together with the methods part of our manuscript provides sufficient detail and overview to ensure that readers can access all experimental information relevant for reproduction.

3) Regarding your data, how is the photon count determined? Could you include a description in the Methods section to explain how imaging signals are converted into photon counts?

Our STED imaging setup employs Avalanche photodiodes, which directly report accumulated photon counts as intensity units in the final image. This was described in the methods part: „Reported pixel intensities are photon counts detected by avalanche photodiode detectors without further processing“.

We added the following statement to the Methods section (lines 610-613):

“The intensity calibration workflow that was demonstrated here (Figure 6) is nonetheless applicable to a variety of setups that report arbitrary intensity units, as long as the photon-to-brightness response is linear, i.e. detectors are not reaching saturated conditions. This can be evaluated based on the linear fit as shown in Figure 6C”.

4) On Line 158, is this intended to be a subtitle?

Yes, thank you for pointing out this error. We corrected it and the following subtitles 2.2.1 and 2.2.2 accordingly.

5) On Line 162, there is additional spacing before and after “single isolated nuclei of.”

Thank you, now corrected.

6) On Line 227, I do not think an additional subtitle here is necessary

We appreciate the reviewer’s comment. However, we have decided to retain and correct this subtitle. It serves to highlight a new variant of our workflow, providing additional scientific insights and expanding the possibilities through the combination of IF and FISH approaches.

7) Lines 371-377 feature a different font color.

Thank you, now corrected.

Reviewer 2 Report

Comments and Suggestions for Authors

Pochechueva et al describe a workflow for Q-FISH microscopy (FISH-STED) using both telomeric and sub-telomeric probes to measure telomere signal intensity on chromosome arm 13q in HeLa 1.3 and cardiac tissue sections. The study was well described and designed.

I only have one major concern. The author’s break down the cell cycle phases into both mitotic and interphase. While the mitotic phase was further broken down into Metaphase, Anaphase, and telophase, Interphase was not broken down into G1, S, and G2. This could have effects on telomere length measurement, due to the synthesis of new telomeric DNA. The authors attempt to reduce the potential impact of this by stating that all interphase cells were in G1, which is unlikely, particularly for HeLa 1.3. Relying solely on telomeric count to determine G1 and G2 is also difficult in cancer cell lines, as each cell often has widely different ploidy. The authors should discuss this in more detail or perform 3D-STED-DyMIN Q-FISH on cells labelled with CDT1 (G1), Geminin (S), and CENP-F (G2).

Author Response

I only have one major concern. The author’s break down the cell cycle phases into both mitotic and interphase. While the mitotic phase was further broken down into Metaphase, Anaphase, and telophase, Interphase was not broken down into G1, S, and G2. This could have effects on telomere length measurement, due to the synthesis of new telomeric DNA. The authors attempt to reduce the potential impact of this by stating that all interphase cells were in G1, which is unlikely, particularly for HeLa 1.3. Relying solely on telomeric count to determine G1 and G2 is also difficult in cancer cell lines, as each cell often has widely different ploidy. The authors should discuss this in more detail or perform 3D-STED-DyMIN Q-FISH on cells labelled with CDT1 (G1), Geminin (S), and CENP-F (G2).

Thank you for the suggestions. We agree that we cannot assert with certainty that all interphase HeLa 1.3 cells in our study were in the G1 phase, as we did not synchronize the cells. Since we have relied on the sc35 marker for M-phase characterization, it was not possible to discern these different interphase stages in the same sample. Following your suggestion, we discussed this limitation in more detail in the Discussion section as follows (lines 406-435):

“Our findings suggest that the analyzed interphase cells were likely in the G1 phase. In contrast, during the G2 phase, doubled chromatids appear, ultimately increasing telomere counts. Importantly, the consistent doubling of spot counts in the M-phase confirms the robustness of the telomere detection and argues against spot merging due spatial crowding or association. Our analysis revealed a homologous population of interphase nuclei with 90 ± 8 telomeres detected by 3D FISH-STED (mean ± SD).  Hence, our analysis did not indicate widely different ploidy in HeLa 1.3 cells, with a consistent doubling of spot counts in M-phase (n = 207 ± 19), showing low relative variance between cells in both groups (CV < 10%). Therefore we can rule out variable polyploidy as a source of error in this study. Another potential source of error could be a large fraction of interphase nuclei in G2, where telomere length measurements and telomere counts might be affected by ongoing synthesis of new telomeric DNA. As noted above, we detected smaller variability of telomere counts in S versus M phase, arguing against a relevant fraction of nuclei in G2. In parallel we detected nearly equal telomere-sizes (Fig. 4D) in S and M phase, arguing against the presence of potentially decompacted enlarged G2 telomeres during synthesis. This is further supported by poor correlation of telomere volumes to signal density (Fig S2), while high correlation would reflect better probe binding during decompaction below. In addition, Deplanche et al. 2015 [47] showed that asynchronous HeLa cells are 79% in G1-, 17% in S, and only 14% in G2/M-phase. This might equally hold true for HeLa 1.3, though it has not been studied to our knowledge. Whether telomere elongation is a sufficient prerequisite for elongated G2 phases is likewise unknown. Therefore, we believe we would have distinguished G1 versus G2 phase nuclei.”

Finally, thank you for the valuable suggestion to enhance our research with more precision and details, including labeling HeLa cells with CDT1, Geminin, and CENP-F. It is indeed an interesting suggestion for further research, yet in our perception beyond the scope of this manuscript, in consideration of the complex encompassing workflow.

Deplanche M, Filho RA, Alekseeva L, Ladier E, Jardin J, Henry G, Azevedo V, Miyoshi A, Beraud L, Laurent F, Lina G, Vandenesch F, Steghens JP, Le Loir Y, Otto M, Götz F, Berkova N. Phenol-soluble modulin α induces G2/M phase transition delay in eukaryotic HeLa cells. FASEB J. 2015 May; 29(5):1950-9. doi: 10.1096/fj.14-260513. Epub 2015 Feb 3. PMID: 25648996; PMCID: PMC4771068.

Reviewer 3 Report

Comments and Suggestions for Authors

Pochechueva et al. have used STED and scanning to visualize and quantify telomeres. The paper is exciting and very clear. I have just minor suggestions:

Introduction:

Line 70-72: topic limitations of Q-FISH: “In part, this can be attributed to telomere analysis with single but not multiple focal imaging planes, leading to low telomere detection rates and subsequent statistical inaccuracy.” Are you taking into account studies using 3D FISH? Such as published in PNAS 2005 (https://doi.org/10.1073/pnas.0407512102)

Line 143; the name FISH-STED was misspelled in this sentence.

Author Response

1) Introduction:

Line 70-72: topic limitations of Q-FISH: “In part, this can be attributed to telomere analysis with single but not multiple focal imaging planes, leading to low telomere detection rates and subsequent statistical inaccuracy.” Are you taking into account studies using 3D FISH? Such as published in PNAS 2005 (https://doi.org/10.1073/pnas.0407512102)

Thank you for highlighting this publication. Indeed, some studies have previously used 3D FISH. We had not explicitly addressed this, since we aim to generally refer to a majority of Q-FISH studies experiencing these limitations. To clarify this point we have modified the Introduction as follows (Lines 78-84):

“Some Q-FISH studies have demonstrated three-dimensional telomere imaging and thus greatly enhanced detection capability (Louis et al., 2005). However, they have relied on diffraction-limited imaging schemes, significantly limiting applicability to more challenging samples, since closely appositional telomeres would likely remain unresolved (recently highlighted by Adam et al., 2019). Despite the application of deconvolution procedures in these studies, the axial sectioning capability has remained an additional limit for attaining higher detection efficiencies.”

We estimate that conventional 3D-deconvolution approaches might reach 200 nm lateral and 500 – 1000 µm axial resolution, compared to 60 nm and 400 nm in our study.

Louis SF, Vermolen BJ, Garini Y, Young IT, Guffei A, Lichtensztejn Z, Kuttler F, Chuang TC, Moshir S, Mougey V, Chuang AY, Kerr PD, Fest T, Boukamp P, Mai S. c-Myc induces chromosomal rearrangements through telomere and chromosome remodeling in the interphase nucleus. Proc Natl Acad Sci U S A. 2005 Jul 5;102(27):9613-8. doi: 10.1073/pnas.0407512102. Epub 2005 Jun 27. PMID: 15983382; PMCID: PMC1172233.

Adam N, Degelman E, Briggs S, Wazen RM, Colarusso P, Riabowol K, Beattie T. Telomere analysis using 3D fluorescence microscopy suggests mammalian telomere clustering in hTERT-immortalized Hs68 fibroblasts. Commun Biol. 2019 Dec 4;2:451. doi: 10.1038/s42003-019-0692-z. PMID: 31815205; PMCID: PMC6893014.

2) Line 143; the name FISH-STED was misspelled in this sentence.

Thank you, now corrected.